# An Improved Extensive Cancellation Method for Clutter Removal in Passive Bistatic Radar

**DOI:** 10.3390/s25216748

**Published:** 2025-11-04

**Authors:** Gang Chen, Siyuan Su, Dandan Zhang, Sujun Wang, Yifan Ping, Fu Li

**Affiliations:** 1China Academy of Space Technology; Xi’an 710100, China; zhangdandan129@163.com (D.Z.); wangsujun504@163.com (S.W.); pingyf1982@126.com (Y.P.); lifu918@sina.com (F.L.); 2Inspur Electronic Information Industry Co., Ltd., Jinan 250101, China; susiyuan@inspur.com

**Keywords:** passive bistatic radar, extensive cancellation algorithm, clutter cancellation

## Abstract

Passive bistatic radar experiences serious clutter echo interference problems; the target echo is submerged by the sidelobes of the strong clutter echoes. Extensive cancellation algorithm is an efficient method for clutter cancellation, but it requires high-order matrix inversion which poses a great challenge to the existing hardware performance and is even impossible to achieve. Aiming at this problem, a fast clutter cancellation method based on the extensive cancellation algorithm is proposed in this paper. In this novel method, the high-order clutter delay matrix is divided into several low-order matrices, and at the same time, multiple sub-matrices are utilized for clutter cancellation simultaneously, which significantly reduces the computational complexity. Simulation results and applications on real data illustrate that the proposed method ensures the clutter cancellation performance while reducing the computational complexity in the passive bistatic radar system.

## 1. Introduction

As a special form of bistatic radar, passive bistatic radar has many advantages over the traditional radar. With no cooperative transmitter, PBR works silently but using the existing broadcasters as radar sources for target detection, location and tracking [1,2]. Due to the silent and bistatic work pattern, PBR is not easily attacked by the anti-radiation missile or disturbed by the hostile radar jammer. In addition, the low-frequency band and the low-altitude coverage characteristic of the broadcasters make the system capable of stealth and low-altitude target detection. The commonly used broadcasters include Frequency Modulation (FM) [3,4,5], Analog Television (ATV) [6,7], Digital Audio Broadcast (DAB) [8,9], Digital Terrestrial Multimedia Broadcast (DTMB) [10,11] and so on. The development trend of the illuminator of broadcaster is from narrowband to broadband.

Since the illuminators of broadcaster usually belong to continuous waves with low transmitting power, signal return from the illuminators to targets are usually very weak [12]. The main lobe of the target echo is usually submerged by the sidelobe of the strong clutter echoes, including the direct-path and multipath signals [13]. Clutter cancellation algorithm is applied to suppress the clutter echoes and then detect the weak target echo. The commonly used clutter cancellation algorithm includes Least Mean Square (LMS) [14,15], Recursive Least Square (RLS) [16,17] and Extensive Cancellation Algorithm (ECA) [18,19,20]. LMS and RLS belong to a closed-loop feedback adaptive filtering algorithm, which obtains the optimal clutter cancellation result through cyclic iteration [21]. In practical applications, especially in highly parallel real-time processing systems, cyclic iteration of the two algorithms will reduce the response speed of the hardware system. ECA belongs to an open-loop algorithm and does not require cyclic iteration, which is especially suitable for parallel processing [22].

However, with the development of the communication technology [23], the signal bandwidth of the illuminator of broadcaster becomes wider and wider, which results in a higher range resolution of the PBR system. Thus, a higher clutter cancellation order is required in the PBR system compared with the narrow band based on the PBR system. High cancellation order in the ECA method means a high-dimensional matrix inversion. It will pose a great challenge to the existing hardware performance and is even impossible to achieve.

To solve the problem of high computational complexity in the ECA method, a batch version of the ECA (ECA-B) [24,25,26] method is proposed. The ECA-B method is achieved by segmenting the signal in time domain for signal processing, meaning the computational complexity could be reduced. This segmentation method does not fundamentally change the overall computational complexity of the algorithm, but it allows for parallel processing, greatly improving the computational efficiency of ECA-B and making it suitable for long time signal processing. However, the clutter cancellation order keeps unchanged compared with the conventional ECA method in this method. The problem of high-order matrix inversion has not been solved. In addition, the higher the segment number is, the worse the cancellation performance will be.

To lower the computational cost in the ECA method, and at the same time maintain the clutter cancellation performance, division order version of the extensive cancellation algorithm (ECA-DO) is proposed in this paper. In this novel method, the clutter delay matrix is divided into several sub-matrices; multiple sub-matrices are utilized for clutter cancellation simultaneously, which ensures the cancellation performance while reducing the inverse order of the matrix. The high-order matrix inversion is transformed into multiple low-order matrix inversions, which is suitable for parallel processing and will lower the operation time.

The rest of this paper is organized as follows. Section 2 introduces the signal model, reviews the related work and details the proposed method. Section 3 analyzes the computational complexity and provides simulation and application on real data results of the method. Section 4 concludes this paper.

## 2. Methods

### 2.1. Construction of Signal Model

A typical passive bistatic radar system is illustrated in Figure 1.

From Figure 1, it could be seen that two receiving channels, the reference channel and surveillance channel, are required. The reference channel is set to receive the direct-path signal from the external illuminator, which is used for clutter cancellation and range-Doppler processing. The surveillance channel is set to collect the signal returns from the interest target in the surveillance area, and it is usually polluted with the direct-path and multipath clutter echoes. Due to the short processing time of the radar system, the reflection intensity of clutter such as ground objects could be considered constant over the processing time. Therefore, the case of stationary clutter is mainly considered in the paper. Thus, the signal in the surveillance channel after down conversion in a digital receiving system with a sampling frequency fs could be represented as(1)$$s_{\mathrm{sur}}\left[n\right]=\sum_{c=1}^{N_{c}}A_{c}d\left[n-\tau_{c}/f_{s}\right]\exp\left[-j2\pi f_{c}n/f_{s}\right]+\sum_{r=1}^{N_{r}}A_{r}d\left[n-\tau_{r}/f_{s}\right]\exp\left[-j2\pi f_{r}n/f_{s}\right]+z_{\mathrm{sur}}\left[n\right]$$
where ***d***[*n*] is the complex envelope of direct-path signal. *A*_c_, *τ*_c_ and *f*_c_ are the complex altitude, the temporal delay and the Doppler frequency of the direct-path and multipath signal in the surveillance channel of the *c*-th delay version. *f*_s_ denotes the sampling frequency. *A_r_*, *τ_r_* and *f_r_* are the complex altitude, temporal delay and the Doppler frequency of the *r*-th target echoes. ***z***_sur_ [*n*] is the thermal noise in the surveillance channel.

Since the direct-path signal is usually received by a directional antenna pointing towards the external illumination, the energy level of the direct-path signal is much stronger than the multipath signal in the reference channel. The contributions of the multipath signal in the reference channel could be ignored. Thus, the signal received by the reference channel after down conversion sampling could be represented as(2)$$s_{\mathrm{ref}}\left[n\right]=Ad\left[n\right]+z_{\mathrm{ref}}\left[n\right]$$
where *A* is the complex altitude of direct-path signal. ***z***_ref_ [*n*] represents the thermal noise in the reference channel.

Usually, the energy level of the target echo is much weaker than the clutter echoes reflected from the ground object. Thus, the clutter echoes in the surveillance channel need to be suppressed before detecting the weak target echo. In the following section, the clutter cancellation method is introduced in detail.

### 2.2. Introduction of the Conventional Processing ECA-B Method

In the ECA method, the temporal delay and the complex altitude of the clutter echoes arr estimated by least square algorithm. Then the estimated clutter echoes are subtracted from the echo signal in the surveillance channel to lower the influence of the clutter sidelobe on target detection. Directly applying the ECA method for clutter cancellation requires a significant number of computational resources and may not even be feasible in the hardware system. To improve the computational efficiency, the ECA-B method has been proposed. The diagram of the algorithm principle is shown in Figure 2.

From Figure 2, it could be seen that both the reference and the surveillance signal are segmented into *B* batches, and the signal length of each batch is *N*/*B*. Each batch of the reference and surveillance signal could be written as(3)$$s_{\mathrm{ref}\_b}\left[n\right]={\left[\begin{array}{cccc} s_{\mathrm{ref}}[\frac{N\left(b-1\right)}{B}]+1 & s_{\mathrm{ref}}[\frac{N\left(b-1\right)}{B}]+2 & \ldots & s_{\mathrm{ref}}[\frac{\mathrm{Nb}}{B}] \end{array}\right]}^{T}$$(4)$$s_{\mathrm{sur}\_b}\left[n\right]={\left[\begin{array}{cccc} s_{\mathrm{sur}}[\frac{N\left(b-1\right)}{B}]+1 & s_{\mathrm{sur}}[\frac{N\left(b-1\right)}{B}]+2 & \ldots & s_{\mathrm{sur}}[\frac{\mathrm{Nb}}{B}] \end{array}\right]}^{T}$$
where (·)^T^ represents the transpose operation.

Thus, the subspace matrix formed by the reference signal and its delay version signal in *b*-th batch could be represented as(5)$$V_{b}\left[n\right]={\left[\begin{array}{cccc} s_{\mathrm{ref}}[\frac{N\left(b-1\right)}{B}]+1 & s_{\mathrm{ref}}[\frac{N\left(b-1\right)}{B}]+2 & \ldots & s_{\mathrm{ref}}[\frac{\mathrm{Nb}}{B}]\\ s_{\mathrm{ref}}[\frac{N\left(b-1\right)}{B}] & s_{\mathrm{ref}}[\frac{N\left(b-1\right)}{B}]+1 & \ldots & s_{\mathrm{ref}}[\frac{\mathrm{Nb}}{B}-1]\\ \vdots & \vdots & \ddots & \vdots \\ s_{\mathrm{ref}}[\frac{N\left(b-1\right)}{B}-K+2] & s_{\mathrm{ref}}[\frac{N\left(b-1\right)}{B}-K+3] & \ldots & s_{\mathrm{ref}}[\frac{\mathrm{Nb}}{B}-K+1] \end{array}\right]}^{T}$$
where *K* is the clutter cancellation order.

In order to achieve clutter cancellation on each signal segment, the following optimization problem requires to be solved.(6)$$\underset{W_{b}}{\min}\left\{{\left\|s_{\mathrm{sur}\_b}\left[n\right]-V_{b}W_{b}\right\|}_{2}^{2}\right\}$$

Equation (6) is a convex optimization problem. The optimal solution could be obtained by solving the partial derivative of ***W****_b_*; it is represented as(7)$$\frac{\partial {\left\|s_{\mathrm{sur}\_b}\left[n\right]-V_{b}W_{b}\right\|}_{2}^{2}}{\partial W_{b}}=2V_{b}^{H}\left(s_{\mathrm{sur}\_b}\left[n\right]-V_{b}W_{b}\right)=0$$
where (·)^H^ represents the conjugate transpose operation.

Solving Equation (7), the optimal solution in the *b*-th batch could be written as(8)$$W_{b}={\left(V_{b}^{H}V_{b}\right)}^{-1}V_{b}^{H}s_{\mathrm{sur}\_b}$$

After clutter cancellation, the residual signal in the *b*-th batch is given by(9)$$e_{b}=s_{\mathrm{sur}\_b}-V_{b}^{H}s_{\mathrm{sur}\_b}=s_{\mathrm{sur}\_b}-V_{b}^{H}{\left(V_{b}^{H}V_{b}\right)}^{-1}V_{b}^{H}s_{\mathrm{sur}\_b}$$

Concatenating the signal after clutter cancellation of each segment, the total residue signal could be represented as(10)$$e={\left[\begin{array}{cccc} e_{1} & e_{2} & \ldots & e_{B} \end{array}\right]}^{T}$$

The ECA-B algorithm does not fundamentally reduce the computational complexity but achieves parallel processing through segmented processing, making it suitable for real engineering. However, the segmented processing method reduces the accuracy of clutter estimation and increases the width of the filter transition band, which is not beneficial for slow and weak target detection. To solve the problem in the ECA-B method, sliding Extensive Cancellation Algorithm (ECA-S) [19] is proposed. The ECA-S method is an improved version of the ECA-B method. In this improved method, the sliding window signal is increased. Thus, there is sufficient processing time for clutter estimation while reducing the filter transition band width, which solves the inconsistency issue between the segment length and the segment number. Essentially, the ECA-S method still belongs to the segmented processing method, and the accuracy of clutter estimation is not as good as the ECA method.

In the two mentioned methods, the order of matrix inversion is not reduced, and the segmented approach will lower the accuracy of clutter estimation, thereby deteriorating the clutter cancellation performance. Thus, it is necessary to design a new algorithm based on ECA to reduce the computation time cost without cancellation performance loss.

### 2.3. Theoretical Derivations of the Proposed ECA-DO Method

In this section, a novel clutter cancellation method-based ECA is proposed. As known, segmented processing belongs to the time-sharing processing method, which will reduce the accuracy of clutter estimation. Thus, it is necessary to consider both segmented processing and simultaneous processing when designing algorithms. From this perspective, the temporal delay of clutter echoes could be divided into several segments, and the temporal delay and complex altitude of the clutter echoes could be estimated on different delay segments. This estimation method is implemented simultaneously. The diagram of the proposed algorithm is shown in Figure 3.

It should be noted that the clutter echoes are usually not stationary in practice; a certain spectral spread in frequency of the clutter echoes occurs due to the wind and other factors. Thus, except static clutter echoes, the spread clutter echoes are also required to be removed from the surveillance signal. Assuming that the multipath signals and spread frequency are distributed within *K* range cell and R Doppler frequency cell, the subspace matrix of clutter cancellation is represented as(11)$$V\left[n\right]={\left[\begin{array}{cccc} s_{\mathrm{ref}}[1] & s_{\mathrm{ref}}[2] & \ldots & s_{\mathrm{ref}}[N]\\ s_{\mathrm{ref}}[1]e^{-2\pi f_{1}n/f_{s}} & s_{\mathrm{ref}}[2]e^{-j2\pi f_{1}n/f_{s}} & \ldots & s_{\mathrm{ref}}[N]e^{-j2\pi f_{1}n/f_{s}}\\ \vdots & \vdots & \ddots & \vdots \\ s_{\mathrm{ref}}[1]e^{-j2\pi f_{R}n/f_{s}} & s_{\mathrm{ref}}[2]e^{-j2\pi f_{R}n/f_{s}} & \ldots & s_{\mathrm{ref}}[N]e^{-j2\pi f_{R}n/f_{s}}\\ \vdots & \vdots & \ddots & \vdots \\ 0 & 0 & \ldots & s_{\mathrm{ref}}[N-K+1]\\ 0 & 0 & \ldots & s_{\mathrm{ref}}[N-K+1]e^{-j2\pi f_{1}n/f_{s}}\\ \vdots & \vdots & \ddots & \vdots \\ 0 & 0 & \ldots & s_{\mathrm{ref}}[N-K+1]e^{-j2\pi f_{R}n/f_{s}} \end{array}\right]}^{T}$$

From Equation (11), it could be seen that the dimension of vector ***V*** is *N* × *KR*. Solving the weight factor of clutter cancellation using vector ***V*** involves finding the inverse of an *KR*-order matrix, which is almost impossible to achieve in practice. Thus, to improve the feasibility of clutter cancellation in practical applications, the inversion of high-order matrices needs to be transformed into the inversion of multiple low-order matrices. However, it is unable to obtain the optimal clutter cancellation weight factor since the integrity of least squares algorithm is desorbed by decomposing the clutter subspace directly for solution. To solve this problem, it is necessary to rewrite the cost function to ensure the integrity of the least squares algorithm while decomposing the clutter subspace. Thus, the subspace matrix of clutter cancellation is decomposed into *P* part, which is represented as(12)$$V\left[n\right]={\left[\begin{array}{cccc} V_{1} & V_{2} & \ldots & V_{P} \end{array}\right]}^{T}$$
where *P* is the total number of decompositions.

The new cost function could be written as(13)$$J=\underset{W_{p}}{\min}\left\{{\left\|s_{\mathrm{sur}}\left[n\right]-\sum_{p}V_{p}W_{p}\right\|}_{2}^{2}\right\}$$
where *p* = 1, 2, …, *P*. ***W****_p_* is the *p*-th part of the optimal solution.

Before solving the cost function above, it is required to prove that Equation (13) is a convex function. Thus, it is required to find the partial derivation of any ***W****_i_*; the result is represented as(14)$$\frac{\partial J}{\partial W_{i}}=-\left(s_{\mathrm{sur}}\left[n\right]-\sum_{p}W_{p}V_{p}\right)V_{i}^{H}$$

Then find partial derivatives for any ***W****_j_*; the result could be represented as(15)$$\frac{\partial^{2}J}{\partial W_{i}\partial W_{j}}=V_{j}V_{i}^{H}$$

Hessian matrix of cost function *J* could be written as(16)$$E=\left[\begin{array}{cccc} V_{1}V_{1}^{H} & V_{1}V_{2}^{H} & \ldots & V_{1}V_{P}^{H}\\ V_{2}V_{1}^{H} & V_{2}V_{2}^{H} & \ldots & V_{2}V_{P}^{H}\\ \vdots & \vdots & \ddots & \vdots \\ V_{P}V_{1}^{H} & V_{P}V_{2}^{H} & \ldots & V_{P}V_{P}^{H} \end{array}\right]$$

Construct a quadratic expression ***x***^H^***Ex***, where ***x*** is an arbitrary non-zero vector, and the vector dimension is 1 × *P*. It could be further represented as(17)$$x^{H}Ex=\sum_{i}\sum_{j}x_{i}^{H}V_{i}{\left(x_{j}^{H}V_{j}\right)}^{H}=x^{H}V{\left(x^{H}V\right)}^{H}={\left(x^{H}V\right)}^{2}$$

It can be seen that the cost function in Equation (17) is a convex function. Find the partial derivative of function *J* with respect to ***W****_i_* and make the derivative equal to zero; it could be represented as(18)$$\left\{\begin{array}{c} \frac{\partial J}{\partial W_{1}}=-\left(s_{\mathrm{sur}}\left[n\right]-\sum_{p}W_{p}V_{p}\right)V_{1}^{H}=0\\ \frac{\partial J}{\partial W_{2}}=-\left(s_{\mathrm{sur}}\left[n\right]-\sum_{p}W_{p}V_{p}\right)V_{2}^{H}=0\\ \vdots \\ \frac{\partial J}{\partial W_{P}}=-\left(s_{\mathrm{sur}}\left[n\right]-\sum_{p}W_{p}V_{p}\right)V_{P}^{H}=0 \end{array}\right.$$

The inversion of the high-order matrix is converted into the inversion of several low-order matrices in the proposed method, essentially, but involving extra multiplication operations. Thus, the computational complexity of the proposed method is not always reduced with the increase in the number of segments in the projection matrix. Assuming the number of segments is set to three for discussion, Equation (18) could be rewritten as(19)$$\left\{\begin{array}{c} \frac{\partial J}{\partial W_{1}}=-\left(s_{\mathrm{sur}}\left[n\right]-W_{1}V_{1}-W_{2}V_{2}-W_{3}V_{3}\right)V_{1}^{H}=0\\ \frac{\partial J}{\partial W_{2}}=-\left(s_{\mathrm{sur}}\left[n\right]-W_{1}V_{1}-W_{2}V_{2}-W_{3}V_{3}\right)V_{2}^{H}=0\\ \frac{\partial J}{\partial W_{3}}=-\left(s_{\mathrm{sur}}\left[n\right]-W_{1}V_{1}-W_{2}V_{2}-W_{3}V_{3}\right)V_{3}^{H}=0 \end{array}\right.$$

Further, Equation (19) could be represented as(20)$$W_{1}V_{1}V_{1}^{H}+W_{2}V_{2}V_{1}^{H}+W_{3}V_{3}V_{1}^{H}=s_{\mathrm{sur}}\left[n\right]V_{1}^{H}$$(21)$$W_{1}V_{1}V_{2}^{H}+W_{2}V_{2}V_{2}^{H}+W_{3}V_{3}V_{2}^{H}=s_{\mathrm{sur}}\left[n\right]V_{2}^{H}$$(22)$$W_{1}V_{1}V_{3}^{H}+W_{2}V_{2}V_{3}^{H}+W_{3}V_{3}V_{3}^{H}=s_{\mathrm{sur}}\left[n\right]V_{3}^{H}$$

Equation (20) could be written as(23)$$W_{1}=\left(s_{\mathrm{sur}}\left[n\right]-W_{2}V_{2}-W_{3}V_{3}\right)V_{1}^{H}{\left(V_{1}V_{1}^{H}\right)}^{-1}$$

Substitute Equation (23) into Equation (21), and ***W***_2_ is represented as(24)$$\begin{array}{ll} W_{2}= & \left[s_{\mathrm{sur}}\left[n\right]\left(V_{2}^{H}-V_{1}^{H}{\left(V_{1}V_{1}^{H}\right)}^{-1}V_{1}V_{2}^{H}\right)-W_{3}\left(V_{3}V_{2}^{H}-V_{3}V_{1}^{H}{\left(V_{1}V_{1}^{H}\right)}^{-1}V_{1}V_{2}^{H}\right)\right]\\ & \cdot {\left(V_{2}V_{2}^{H}-V_{2}V_{1}^{H}{\left(V_{1}V_{1}^{H}\right)}^{-1}V_{1}V_{2}^{H}\right)}^{-1} \end{array}$$

Substitute Equation (23) and (24) into Equation (22), and ***W***_2_ is represented as(25)$$W_{3}=W_{\mathrm{num}}\cdot W_{\mathrm{den}}$$
where ***W***_num_ and ***W***_den_ could be written as(26)$$\begin{array}{ll} W_{\mathrm{num}}= & s_{\mathrm{sur}}\left[n\right]\left[\left(V_{3}^{H}-V_{1}^{H}{\left(V_{1}V_{1}^{H}\right)}^{-1}V_{1}V_{3}^{H}\right)-\left(V_{2}^{H}-V_{1}^{H}{\left(V_{1}V_{1}^{H}\right)}^{-1}V_{1}V_{2}^{H}\right)\right.\\ & \left.\cdot {\left(V_{2}V_{2}^{H}-V_{2}V_{1}^{H}{\left(V_{1}V_{1}^{H}\right)}^{-1}V_{1}V_{2}^{H}\right)}^{-1}\left(V_{2}V_{3}^{H}-V_{2}V_{1}^{H}{\left(V_{1}V_{1}^{H}\right)}^{-1}V_{1}V_{3}^{H}\right)\right] \end{array}$$(27)$$\begin{array}{ll} W_{\mathrm{den}}= & \left[\left(V_{3}V_{3}^{H}-V_{3}V_{1}^{H}{\left(V_{1}V_{1}^{H}\right)}^{-1}V_{1}V_{3}^{H}\right)-\left(V_{3}V_{2}^{H}-V_{3}V_{1}^{H}{\left(V_{1}V_{1}^{H}\right)}^{-1}V_{1}V_{2}^{H}\right)\right.\\ & {\left.\cdot {\left(V_{2}V_{2}^{H}-V_{2}V_{1}^{H}{\left(V_{1}V_{1}^{H}\right)}^{-1}V_{1}V_{2}^{H}\right)}^{-1}\left(V_{2}V_{3}^{H}-V_{2}V_{1}^{H}{\left(V_{1}V_{1}^{H}\right)}^{-1}V_{1}V_{3}^{H}\right)\right]}^{-1} \end{array}$$

After obtaining weight factor ***W***_3_ through the above equation, ***W***_2_ could be obtained by Equation (24). Moreover, ***W***_1_ could also be obtained by Equation (23).

When obtaining the weight factor ***W***_1_, ***W***_2_ and ***W***_3_, the residual signal could be represented as(28)$$e\left[n\right]=s_{\mathrm{sur}}\left[n\right]-V_{1}W_{1}-V_{2}W_{2}-V_{3}W_{3}$$

After clutter cancellation, most of the clutter echo signals are suppressed, but there is still a small amount of clutter residue and thermal noise. The energy level of the weak target echo is still weaker than the thermal noise. Thus, range-Doppler processing between the reference signal and the signal residue after clutter cancellation is required to raise the energy level of the weak target echo and further suppress the interference signal, which is performed as(29)$$S_{\mathrm{mat}}\left(\tau ,f\right)=\sum_{n=1}^{N}s_{\mathrm{ref}}^{*}\left[n-\tau \right]e\left[n\right]\exp\left(-j2\pi fn/N\right)$$
where *τ* and *f* are the temporal delay bin and Doppler frequency bin, respectively. (·) ^*^ is the conjugate operation.

From the solving process, it could be seen that the inversion of the high-order matrix is transformed into the inversion of multiple low-order matrices. The dimension of the inverse matrix involved in the solving process in this method is *KR/P* × *KR/P*, whereas the original solving process is *KR* × *KR*. The computational complexity of matrix inversion is the cube of the matrix order. The computational complexity of matrix inversion of the proposed method in this process is 1/*P*^3^ of the original algorithm, which shows that the proposed method could improve the feasibility of the algorithm.

Although the proposed method involves more multiplication operations, these multiplications are mostly repetitive and could be performed in parallel, which is less than the computation cost of solving the inverse of high-dimensional matrices. A detailed comparison and analysis of the computational complexity of the algorithms mentioned above are discussed in the next section.

## 3. Results

### 3.1. Computational Complexity Analysis

In this section, the computational complexity of ECA, ECA-B, ECA-S and ECA-DO methods is discussed.

#### 3.1.1. ECA Method

The optimal solution of the ECA could be represented as(30)$$W={\left(V^{H}V\right)}^{-1}V^{H}s_{\mathrm{sur}}\left[n\right]$$

The dimension of the ***V*** is *N* × *KR*. The solving process could be represented as the following three sequential steps:

(1)Let $R_{0}=V^{H}V$ and $R_{1}=V^{H}s_{\mathrm{sur}}\left[n\right]$, the total number of complex multiplications required are *N*(*KR*)^2^ and *NKR*. Thus, the computational complexity in this step is *O*[*N*(*KR*)^2^].(2)Let $R_{n}=R_{0}^{-1}$, the computational complexity in this step is *O*[(*KR*)^3^].(3)Let $W=R_{n}R_{1}$, the number of complex multiplications is (*KR*)^2^ and the computational complexity in this step is *O*[(*KR*)^2^].

Thus, the total computational complexity of the ECA method is *O*[*N*(*KR*)^2^] *+ O*[(*KR*)^3^] + *O*[(*KR*)^2^].

#### 3.1.2. ECA-B and ECA-S Method

The optimal solution of the ECA-B algorithm is written as(31)$$W_{b}={\left(V_{b}^{H}V_{b}\right)}^{-1}V_{b}^{H}s_{\mathrm{sur}\_b}$$

The dimension of the ***V****_b_* is *N/B* × *KR*. The solving process could be represented as the following three sequential steps:(1)Let $R_{0b}=V_{b}^{H}V_{b}$ and $R_{1b}=V_{b}^{H}s_{\mathrm{sur}\_b}\left[n\right]$, the total number of complex multiplications required are *N/B·*(*KR*)^2^ and *N/B·KR*. Thus, the computational complexity in this step is *O*[*N/B·*(*KR*)^2^].(2)Let $R_{\mathrm{nb}}=R_{0b}^{-1}$, the computational complexity in this step is *O*[(*KR*)^3^].(3)Let $W_{b}=R_{\mathrm{nb}}R_{1b}$, the number of complex multiplications is (*KR*)^2^ and the computational complexity in this step is *O*[(*KR*)^2^].

Each segment could be computed in parallel. Thus, the total computational complexity of the ECA-B method is *O*[*N/B·*(*KR*)^2^] *+ O*[(*KR*)^3^] + *O*[(*KR*)^2^].

The ECA-S method is an improved version of the ECA-B method. The difference between the two methods is the addition of sliding window signals in clutter estimation. Thus, the total computation of the ECA-S method is *O*[(*N/B* + 2*N_s_*)*·*(*KR*)^2^] *+ O*[(*KR*)^3^] + *O*[(*KR*)^2^], where *N_s_* is the length of the sliding window signal, *N*_s_ < (*N/B*)*/*2.

#### 3.1.3. ECA-DO Method

In order to facilitate the analysis of the proposed method, ***W***_num_ and ***W***_den_ could be written as(32)$$\begin{array}{ll} W_{\mathrm{num}}= & s_{\mathrm{sur}}\left[n\right]\left[\left(V_{3}^{H}-V_{1}^{H}{\left(R_{11}\right)}^{-1}R_{13}\right)-\left(V_{2}^{H}-V_{1}^{H}{\left(R_{11}\right)}^{-1}R_{12}\right)\right.\\ & \left.\cdot {\left(R_{22}-R_{21}{\left(R_{11}\right)}^{-1}R_{12}\right)}^{-1}\left(R_{23}-R_{21}{\left(R_{11}\right)}^{-1}R_{13}\right)\right] \end{array}$$(33)$$\begin{array}{ll} W_{\mathrm{den}}= & \left[\left(R_{33}-R_{31}{\left(R_{11}\right)}^{-1}R_{13}\right)-\left(R_{32}-R_{31}{\left(R_{11}\right)}^{-1}R_{12}\right)\right.\\ & {\left.\cdot {\left(R_{22}-R_{21}{\left(R_{11}\right)}^{-1}R_{12}\right)}^{-1}\left(R_{23}-R_{21}{\left(R_{11}\right)}^{-1}R_{13}\right)\right]}^{-1} \end{array}$$
where $R_{11}=V_{1}V_{1}^{H}$, $R_{12}=V_{1}V_{2}^{H}$, $R_{13}=V_{1}V_{3}^{H}$, $R_{21}=V_{2}V_{1}^{H}$, $R_{22}=V_{2}V_{2}^{H}$, $R_{23}=V_{2}V_{3}^{H}$, $R_{31}=V_{3}V_{1}^{H}$, $R_{32}=V_{3}V_{2}^{H}$, $R_{33}=V_{3}V_{3}^{H}$.

The dimension of the ***V****_p_*(*p* = 1,2,3) is *N* × *KR/*3. The solving process could be represented as the following eight sequential steps:(1)The number of complex multiplications in solving $R_{11}$, $R_{12}$, $R_{13}$, $R_{21}$, $R_{22}$, $R_{23}$, $R_{31}$, $R_{32}$, $R_{33}$ both are *N* × (*KR/*3)^2^. Thus, the computational complexity in this step is *O*[*N*(*KR/*3)^2^].(2)Let $R_{\mathrm{np}}=R_{11}^{-1}$, the computational complexity in this step is *O*[(*KR*/3)^3^].(3)Let $Y_{1}=V_{3}^{H}-V_{1}^{H}R_{\mathrm{np}}R_{13}$, $Y_{2}=V_{2}^{H}-V_{1}^{H}R_{\mathrm{np}}R_{12}$, $Y_{3}=R_{22}-R_{21}R_{\mathrm{np}}R_{12}$, $Y_{4}=R_{23}-R_{21}R_{\mathrm{np}}R_{13}$
$Y_{5}=R_{33}-R_{31}R_{\mathrm{np}}R_{13}$
$Y_{6}=R_{32}-R_{31}R_{\mathrm{np}}R_{12}$
$Y_{7}=R_{22}-R_{21}R_{\mathrm{np}}R_{12}$, $Y_{8}=R_{23}-R_{21}R_{\mathrm{np}}R_{13}$. The total numbers of complex multiplications in solving ***Y***_1_ and ***Y***_2_ are both (*KR*/3)^3^ + *N*(*KR*/3)^2^. The total numbers of complex multiplications in solving ***Y***_3_ to ***Y***_8_ are both 2 × (*KR*/3)^3^. Thus, the computational complexity in this step is *O*[(*KR*/3)^3^ + *N*(*KR*/3)^2^].(4)Let $Y_{n3}=Y_{3}^{-1}$ and $Y_{n7}=Y_{7}^{-1}$, the computational complexity in this step is *O*[(*KR/*3)^3^].(5)Let $Z_{1}=Y_{1}-Y_{2}Y_{n3}Y_{4}$ and $Z_{1}=Y_{5}-Y_{6}Y_{n7}Y_{8}$, the total numbers of complex multiplications in solving ***Z***_1_ and ***Z***_2_ are (*KR*/3)^3^ + *N*(*KR*/3)^2^ and 2 × (*KR*/3)^3^. Thus, the computational complexity in this step is *O*[(*KR*/3)^3^ + *N*(*KR*/3)^2^].(6)$W_{\mathrm{num}}=s_{\mathrm{sur}}\left[n\right]Z_{1}$, the total number of complex multiplications is *NKR*/3. The computational complexity is *O*(*NKR*/3). $W_{\mathrm{den}}=Z_{2}^{-1}$, the computational complexity is *O*[(*KR*/3)^3^]. Thus, the computational complexity in this step is *O*[(*KR*/3)^3^].(7)From the analysis above, ***W***_2_ could be represented as $W_{2}=\left[s_{\mathrm{sur}}\left[n\right]Y_{2}-W_{3}Y_{6}\right]Y_{n3}$ where $s_{\mathrm{sur}}\left[n\right]Y$ is calculated in steps 5–6. The complex multiplications in solving $s_{\mathrm{sur}}\left[n\right]Y$ could be ignored in this step and the total complex multiplications in this step involves $W_{3}Y_{6}Y_{n3}$. Thus, the total computational complexity in this step is *O* [2(*KR*/3)^2^].(8)Similarly, $V_{1}^{H}{\left(V_{1}V_{1}^{H}\right)}^{-1}$ could be calculated in steps 1–3. Thus, the total computational complexity in solving ***W***_1_ is *O* [2*NKR*/3].

Thus, the total computational complexity of the proposed ECA-DO method is *O*[*N*(*KR/*3)^2^] + *O*[(*KR*/3)^3^] + *O*[(*KR*/3)^3^ + *N*(*KR*/3)^2^] *+ O*[(*KR/*3)^3^] + *O*[(*KR*/3)^3^ + *N*(*KR*/3)^2^] + *O*[(*KR*/3)^3^] +*O* [2(*KR*/3)^2^] + *O* [2*NKR*/3].

#### 3.1.4. Computational Complexity Comparison

Next, the computational complexity of the four clutter cancellation algorithms under different scenarios is analyzed, supposing that the system sampling rate and the extended clutter cancellation order are set to 8 MHz and 20, respectively. In this part, the integration time varies from 0.1 s to 1 s, and the clutter cancellation order increases from 1000 to 3000. Corresponding to analysis, *R* is fixed at 20. *K* varies from 100 to 3000 and *N* increases from 8 × 10^5^ to 8 × 10^6^. Figure 4 shows the comparison results of the computational complexity of different algorithms for fixed clutter cancellation order and cancellation data length, respectively.

From Figure 4, it could be seen that the computational complexity of the four methods is increased with the increase in clutter cancellation order and cancellation data length. The complexity of the ECA method is greater than that of ECA-B, ECA-S and ECA-DO in both cases. Due to the addition of the sliding window signal, the computation complexity of the ECA-S method is greater than that of the ECA-B method in both cases. What is more, the computational complexity of the proposed ECA-DO method is slightly lower than that of the ECA-B method. As the order of clutter cancellation increases, the growth rate of the proposed ECA-DO method in computational complexity is smaller than that of the ECA-B method. The proposed ECA-DO method has a significant advantage in computational complexity of high-order cancellation and is especially suitable for current high-resolution passive bistatic radar systems. To further show the advantage of the proposed method, simulation results are presented in the next section.

### 3.2. Simulation Results

In this section, DTMB signal is adopted as the radar source in the simulation. DTMB signal adopts Quadrature Amplitude Modulation (QAM) and Time-Domain Synchronous Orthogonal Frequency Division Multiplexing (TDS-OFDM) as its modulation and multiple access method. The bandwidth of DTMB signal is 7.56 MHz, and the sampling rate of the PBR system is 8 MHz. The total integration time is 0.1 s. The waveform and the spectrum of the DTMB signal are shown in Figure 5.

Supposing that five clutter echoes and three targets are received by the surveillance antenna, the specific simulation parameters are listed in Table 1. In addition, the total clutter cancellation order is 500. In the ECA-B and ECA-S methods, the segment number and the sliding window signal length are 10 and 10,000, respectively. *R* is fixed at 20. The processor of the computer is Inter(R) Core (TM) Ultra 9 185 H 2.30 GHz, with 32 GB of memory.

According to the parameter above, the ECA, ECA-B, ECA-S and ECA-DO methods are applied for clutter cancellation. Range-Doppler processing results are shown in Figure 6.

It could be seen from Figure 6a that the main component of the surveillance channel is clutter echoes including direct-path and multipath signals. The main lobe of the weak target echo is submerged by the sidelobe of the clutter echoes, which could not be detected. The noise platform is about −15 dB. The peak in the red circle is the ambiguity sidelobe of the direct-path signal, which is caused by the repeated pilot in DTMB signals. From Figure 6, it is observed that the clutter echoes are suppressed by applying the ECA, ECA-B, ECA-S and the proposed ECA-DO method. The main lobe of the three target echoes could be detected. To further demonstrate the performance of the methods, Doppler and range channels of the third target are displayed for discussion.

From Figure 7, it could be seen that the platform of the four results is −51 dB, −49 dB, −50 dB and −51 dB. It could be calculated that the cancellation ratios of the four methods are 35 dB, 32 dB, 34 dB and 35 dB, respectively. The main reason is that the ECA methods are based on the principle of total LS for clutter estimation, whereas the estimation performance of the ECA-B and ECA-S methods in each segment are worse than that of the total cancellation, such as in the ECA and ECA-DO methods. Figure 8 shows the relationship between the data length and cancellation ratio of the three methods.

From Figure 8, it could be seen that the cancellation ratios of the ECA-B and ECA-S methods increased with the increase in cancellation data length, whereas the cancellation ratio of ECA and ECA-DO methods remains unchanged with the increase in cancellation data length. It should be noted that the cancellation ratio curve of the ECA method and ECA-DO method are coincident since the ECA-DO method is a fast implementation method of ECA. The computational complexity of different methods is shown in Table 2. The computational cost of the ECA-DO method is significantly lower than ECA method and is slightly lower than ECA-B and ECA-S method, which illustrates the effectiveness of the proposed method.

### 3.3. Applications on Real Data

In this part, real data collected from DTMB based on passive bistatic radar is employed to further illustrate the effectiveness of the proposed method. Figure 9 shows the position of the PBR receiver and the DTMB transmitter in Shaanxi, China. The DTMB transmitter in Xi’an is adopted as the illuminator of opportunity, whereas the PBR receiver is located in Xianyang. The target to be observed is an unmanned aerial vehicle (UAV). The UAV and control equipment are shown in Figure 10. The carrier frequency and signal bandwidth of the DTMB transmitter are 706 MHz and 7.56 MHz, respectively. The signal received from antenna are down-converted to baseband and sampled by an A/D converter with a sampling rate of 8 MHz. The observing time of the system is 0.1 s. ECA, ECA-B, ECA-S and the proposed ECA-DO method are employed for clutter cancellation; the processing results are shown in Figure 11.

It could be seen that from Figure 11a that the main component of the surveillance channel is the direct-path signal and multipath signal, involving zero-frequency and non zero-frequency clutter echoes. The main lobe of the weak target echoes is submerged by the sidelobes of the strong clutter echoes. From Figure 11b, it could be seen that most of the clutter echoes are suppressed effectively by the ECA method, and multiple peaks of the target could be detected through range-Doppler processing. Although the clutter echoes could be suppressed by the ECA-B and ECA-S method in Figure 11c,d, there is residual clutter due to the influence of segmentation, and the target amplitude is also affected. Some targets even cannot be detected. From Figure 11e, it could be seen that most of the targets could be removed from the surveillance channel, and the main lobe of the target could be detected by the ECA-DO method. The clutter cancellation performance of the proposed method is same as that of the ECA method, which illustrates the effectiveness of the proposed method.

To further examine the proposed ECA-DO method, 100 frames of continuous real data are employed. Range-Doppler results and constant false alarm rate (CFAR) detection results obtained by using the ECA-DO method are shown in Figure 12. In CFAR processing, cell-averaging CFAR (CA-CFAR) detection method is adopted to extract the parameter of the moving target. The CFAR detection threshold is set to 3.5. From Figure 12, it could be seen that several target motion trajectories, including UAV, are detected. As shown in Figure 13, the extracted point results are consistent with the UAV motion trajectory results displayed on UAV control device, which verifies the cancellation performance of the proposed ECA-DO method.

### 3.4. Discussion

With the development of communication technology, the signal bandwidth of the communication illuminator becomes wider and wider. Wideband signal enhances the range resolution of the PBR system at the same time, resulting in increased computational load, especially in clutter cancellation processing. The main idea of this paper is to propose a fast ECA method to reduce computational burden. From computational complexity analysis, the proposed method has significant advantages in computational complexity over the conventional ECA method and is better than the ECA-B and ECA-S methods. From the simulation results, it could be seen that the proposed method is theoretically equivalent to ECA and the performance of the proposed method is better than ECA-B and ECA-S. Simulation results shows that the proposed method ensures the cancellation performance while reducing the computational complexity. Applications on real data also illustrates that the controlled UAV target is successfully detected and a continuous trajectory is formed by utilizing the ECA-DO method conforming the effectiveness of the proposed method.

## 4. Conclusions

To solve the high computational complexity problems in the ECA method, the ECA-DO method is proposed in the paper. In this novel clutter cancellation method, the clutter delay matrix is divided into several sub-matrices; multiple sub-matrices are utilized for clutter cancellation simultaneously, which ensures the cancellation performance while reducing the inverse order of the matrix. Specifically, the high-order clutter delay estimation matrix inversion is transformed into multiple low-order matrices inversion in the first step. Then the optimal weight factors of several low-order matrices of clutter delay estimation are solved by the newly constructed cost function simultaneously, which ensures the integrity of clutter cancellation. The processing flow could be implemented in parallel without performance loss compared with the ECA method. Computational complexity analysis and simulation results illustrate that the computational complexity in clutter cancellation could be reduced while ensuring clutter cancellation performance by applying the proposed ECA-DO method.

In addition, real data collected from a DTMB based on passive bistatic radar also supports the proposed method. In reality, there are more types of clutter echoes, and the cancellation order is much higher. It is almost impossible to perform high-order matrix inversion in practice. Thus, it is essential to lower the order of the matrix inversion to improves the real-time capability of passive bistatic radar systems. In future work, we will focus on achieving the algorithm on the platform suitable for parallel processing to further illustrate the effectiveness of the proposed method.

## Figures and Tables

**Figure 1 sensors-25-06748-f001:**
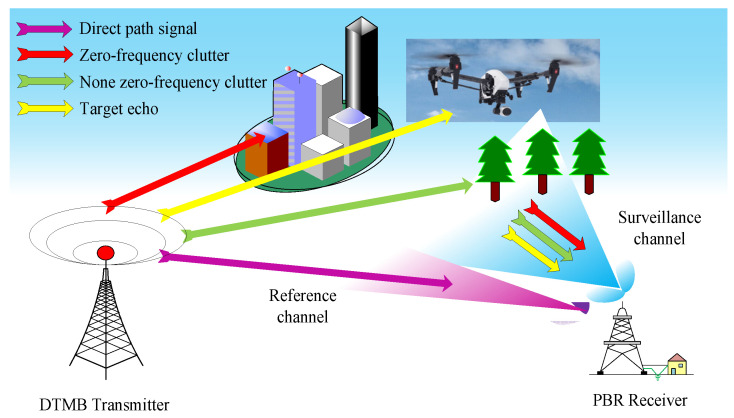
PBR geometry.

**Figure 2 sensors-25-06748-f002:**
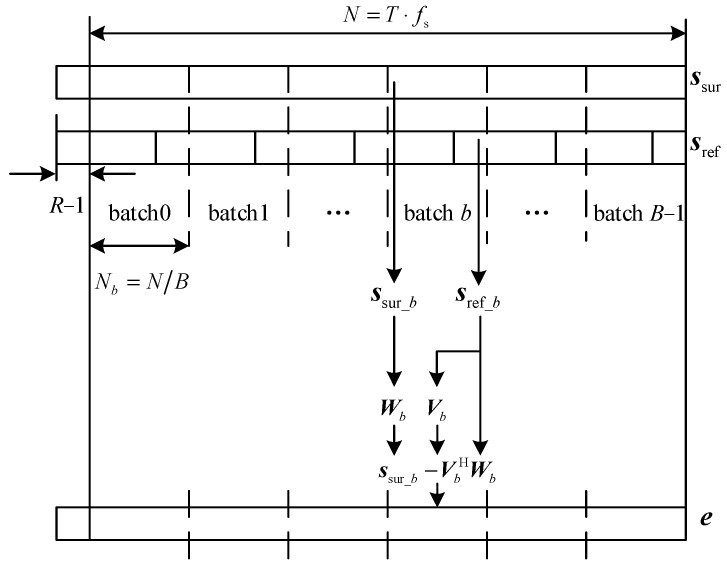
Diagram of ECA-B.

**Figure 3 sensors-25-06748-f003:**
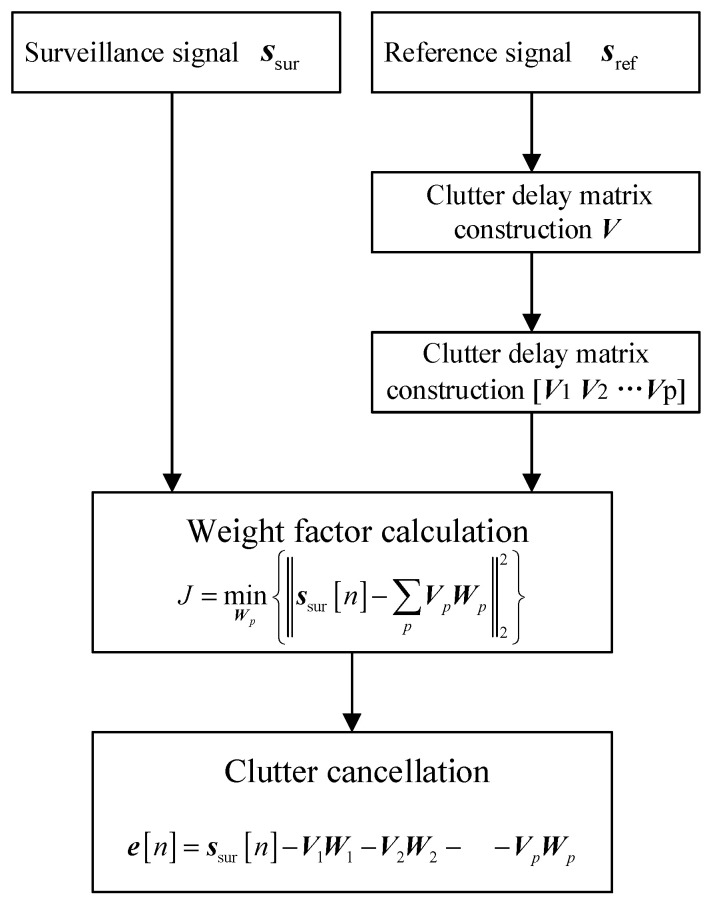
The diagram of ECA-DO.

**Figure 4 sensors-25-06748-f004:**
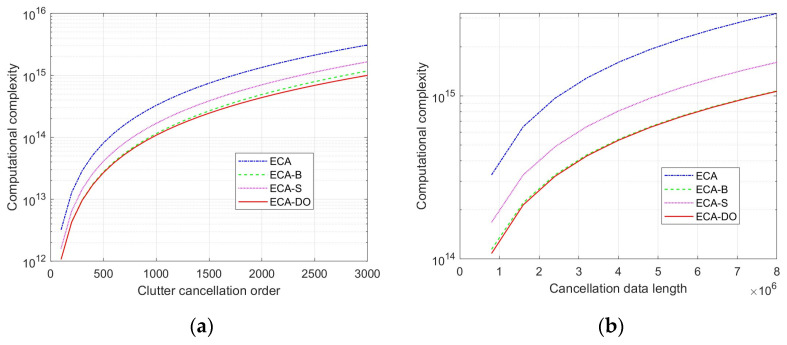
Computational complexity analysis: (**a**) relationship between clutter cancellation order and computational complexity; (**b**) relationship between cancellation data length and computational complexity.

**Figure 5 sensors-25-06748-f005:**
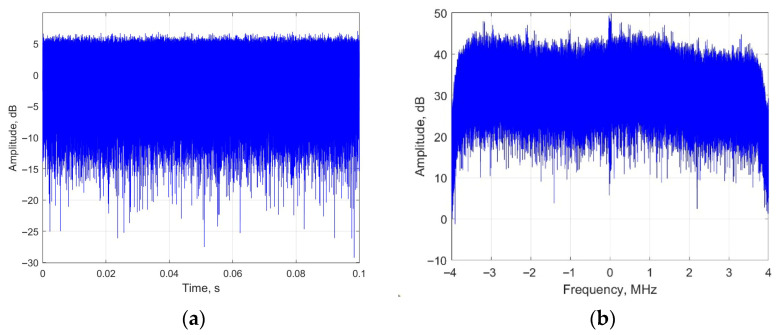
Waveform and spectrum of DTMB signal: (**a**) waveform; (**b**) spectrum.

**Figure 6 sensors-25-06748-f006:**
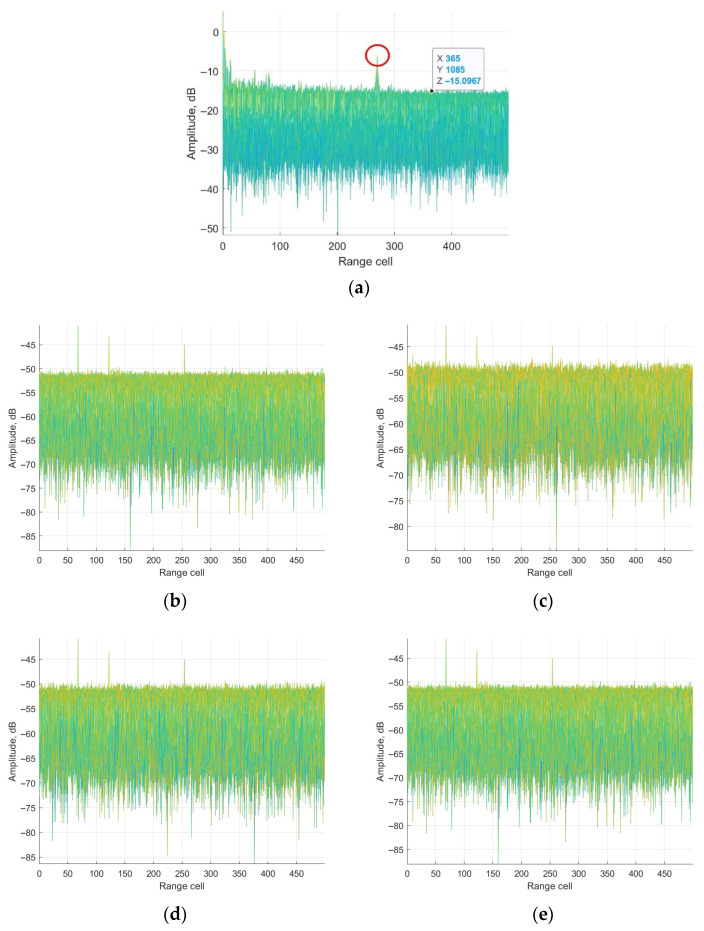
Range-Doppler results in range dimension: (**a**) without cancellation; (**b**) ECA method; (**c**) ECA-B method; (**d**) ECA-S method; (**e**) ECA-DO method.

**Figure 7 sensors-25-06748-f007:**
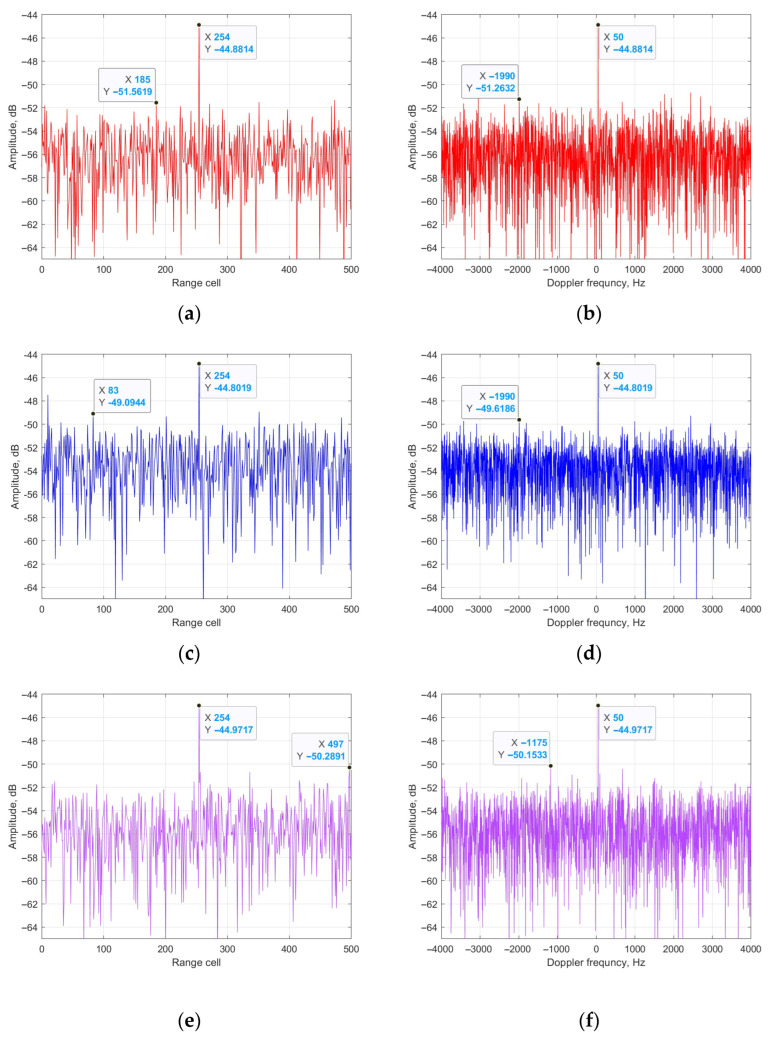
Output of four methods: (**a**) zero-Doppler cut of ECA; (**b**) zero-range cut of ECA; (**c**) zero-Doppler cut of ECA-B; (**d**) zero-range cut of ECA-B; (**e**) zero-Doppler cut of ECA-S; (**f**) zero-range cut of ECA-S; (**g**) zero-Doppler cut of ECA-DO; (**h**) zero-range cut of ECA-DO.

**Figure 8 sensors-25-06748-f008:**
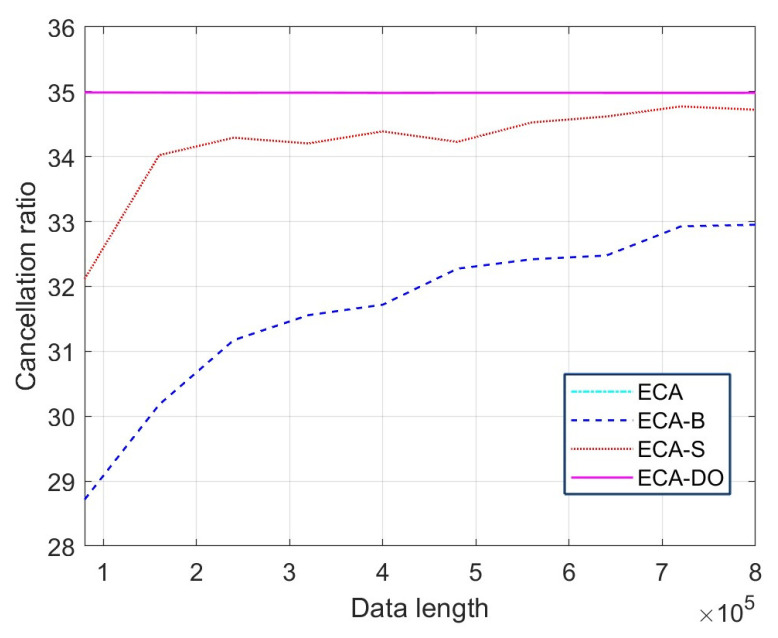
The variation in cancellation ratio with cancellation length.

**Figure 9 sensors-25-06748-f009:**
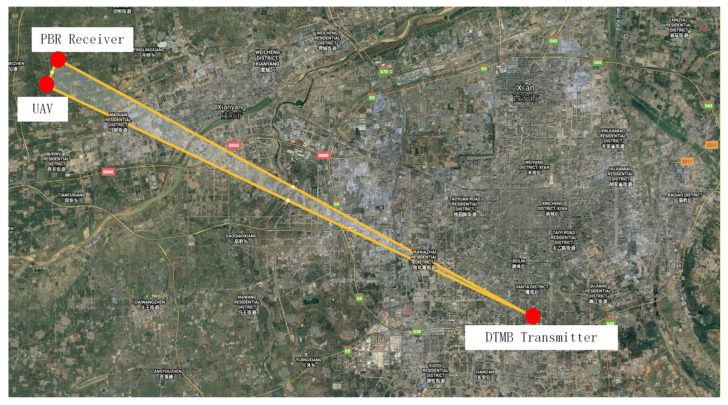
Position of PBR system.

**Figure 10 sensors-25-06748-f010:**
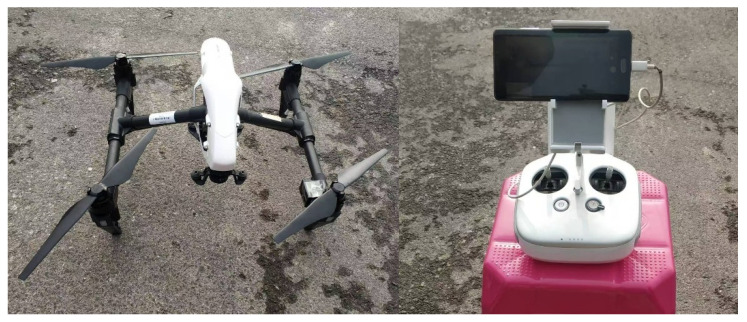
UAV and control equipment.

**Figure 11 sensors-25-06748-f011:**
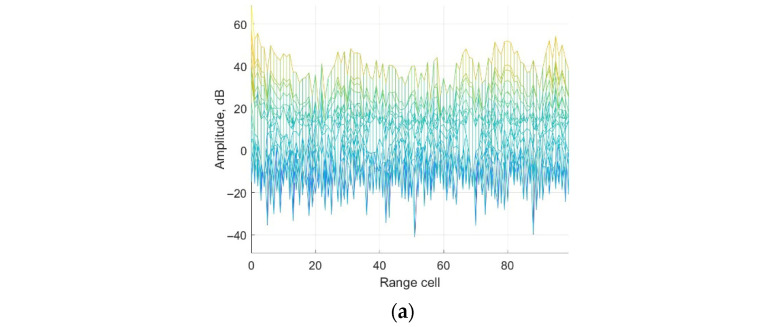
Range-Doppler result in range dimension on real data: (**a**) without cancellation; (**b**) ECA method; (**c**) ECA-B method; (**d**) ECA-S method; (**e**) ECA-DO method.

**Figure 12 sensors-25-06748-f012:**
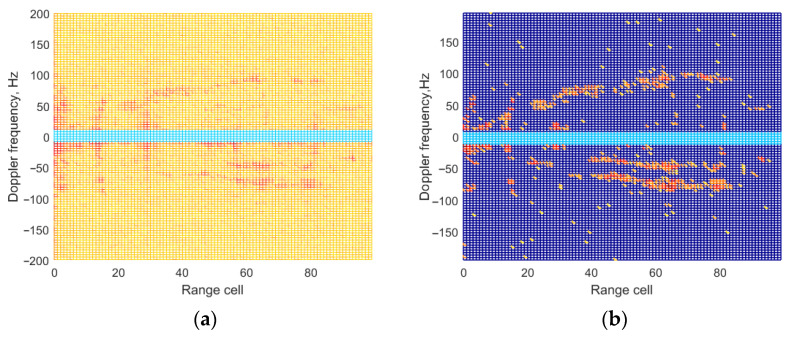
Result on real data with 100 frames using ECA-DO method: (**a**) Range-Doppler result; (**b**) CFAR result.

**Figure 13 sensors-25-06748-f013:**
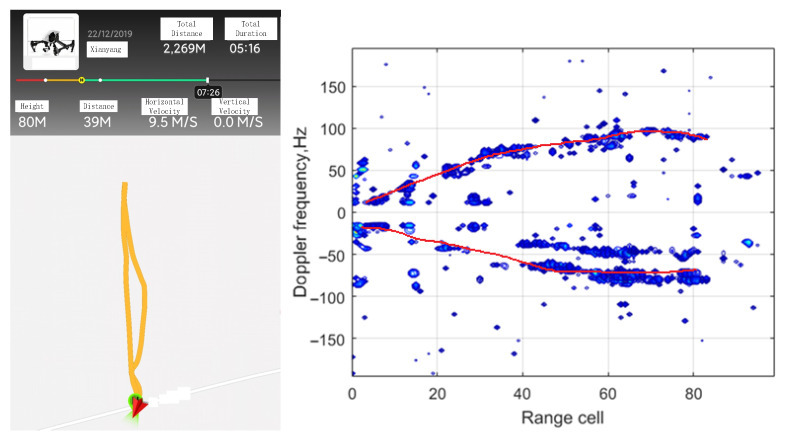
Comparison between plot and actual trajectory.

**Table 1 sensors-25-06748-t001:** Simulation parameters.

Signal Type	SNR (dB)	Range Cell	Doppler Frequency (Hz)
Direct path signal	35	0	0
Zero-frequency Multipath 1	20	10	0
Zero-frequency Multipath 2	15	45	0
Non-zero frequency multipath 1	10	6	−10
Non-zero frequency multipath 2	8	3	20
Target 1	−11	68	2212100
Target 2	−13	122	120
Target 3	−15	254	−50

**Table 2 sensors-25-06748-t002:** Computational complexity of the four methods.

Method	ECA	ECA-B	ECA-S	ECA-DO
Number of complexity multiplications	8.1000 × 10^13^	2.7667 × 10^13^	4.1000 × 10^13^	2.6857 × 10^13^

## Data Availability

The data presented in this study are available on request from the corresponding author.

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
