# Peer review of "An Improved Extensive Cancellation Method for Clutter Removal in Passive Bistatic Radar"

_sensors, 2025, doi:10.3390/s25216748_

Round 1

Reviewer 1 Report

Comments and Suggestions for Authors

The manuscript lacks a clear comparison with the Sliding ECA, Subspace Projection, or GPU-parallel ECA methods.

Theoretical complexity reduction claims (to 1/P²) are not experimentally verified.

No quantitative time or memory benchmark for the proposed method.

A short algorithm block for ECA-DO would make implementation straightforward.

Try different SNRs, clutter levels, and number of targets.

Provide computational setup — CPU type, RAM, programming environment.

Provide measured runtime (in seconds) or FLOP count for ECA, ECA-B, and ECA-DO.

Reviewer 2 Report

Comments and Suggestions for Authors

This paper presents a division-order extensive cancellation algorithm (ECA-DO) for clutter suppression in passive bistatic radar systems. The method divides the high-order clutter delay subspace into several lower-order subspaces, achieving similar suppression performance as the traditional ECA while significantly reducing computational complexity. I have some comments about the paper.

  1. The novelty could be emphasized more clearly by comparing with ECA-B and other low-complexity ECA variants.
  2. Include one paragraph on sensitivity to clutter non-stationarity and model mismatch (e.g., spread in Doppler bins, DTMB pilots), and state any limits where ECA-DO may degrade..
  3. Provide a concise replication checklist (key 𝐾,𝑅 batch/segment settings, DTMB parameters, CFAR settings) to ease comparison by other groups.
  4. Minor polishing of figures, symbols, and formatting would improve readability.
  5. It may be valuable to include recent related works that also follow a model-driven optimization framework. For instance, the paper ‘Enhanced Channel Estimation for Hybrid-Field XL-MIMO Systems Using Joint Sparse Bayesian Learning’ illustrates how integrating physical modeling with statistical learning can improve robustness and efficiency—conceptually related to your structure-aware approach.

Reviewer 3 Report

Comments and Suggestions for Authors

This paper presents a novel extensive cancellation method for clutter removal in passive bistatic radar. The idea is interesting, the proposed method is supported by simulations and application on real data. I have some questions, if addressed would strengthen the paper.

  1. How does the proposed algorithm reduce computational complexity?
  2. Is the cascaded processing of clutter cancellation algorithms feasible?
  3. The cancellation ratio of ECA method and the proposed algorithm completely overlaps. Please explain the reason.
  4. The total integration time is 0.1s, whereas the time in figure 5a is 0.2s. it requires to be corrected.
  5. The curve of ECA method cannot be seen in Figure 8, please correct it.

How to associate the points in Figure 13 with the actual motion trajectory of the target?

Round 2

Reviewer 1 Report

Comments and Suggestions for Authors

thanks for considering the comments.